Automated Knowledge Base Construction (2021)              Conference paper

# A Case Study in Bootstrapping Ontology Graphs from Textbooks

**Vinay K. Chaudhri**                    VINAYC@STANFORD.EDU
**Matthew Boggess**                    MATTBOGG@STANFORD.EDU
**Han L. Aung**                    HANLAUNG@CS.STANFORD.EDU
*Department of Computer Science, Stanford University, Stanford, CA, 94305, USA*

**Debshila B. Mallick**                    DEBSHILA@RICE.EDU
**Andrew C. Waters**                 ANDREW.E.WATERS@GMAIL.COM
**Richard E. Baraniuk**                    RICHB@RICE.EDU
*OpenStax, Rice University, Houston, TX, 77005*

## Abstract

Ontology graphs are graphs in which the nodes are generic classes and edges have labels that specify the relationships between the classes. In this paper, we address the question: to what extent can automated extraction and crowd sourcing techniques be combined to bootstrap the creation of comprehensive and accurate ontology graphs? By adapting the state-of-the-art language model BERT to this task, and by leveraging a novel crowd sourced relationship selection task, we show that we can use this approach to bootstrap the ontology graph creation for further refinement and improvement through human effort.

## 1. Introduction

Ontology learning from text is the problem of extracting classes, relationships, and their complete logical definitions [12]. An ontology graph is a subset of an ontology in that it focuses primarily on representing classes and class to class relationships. For example, given the sentence: "*Cytoplasm consists of jelly-like cytosol within a cell*", its ontology graph contains nodes for *cytoplasm, cytosol* and *cell*, with an edge between *cytoplasm* and *cytosol* labeled as *material*, and an edge between *cell* and *cytoplasm* labelled as *is inside*. In addition, it contains a node *jelly-like* connected to *cytosol* using the label *viscosity*. The relationships such as *material, is inside* and *viscosity* are not directly mentioned in the text, and there could be many different ways to express these relationships in text.

Ontology graphs are different from fact graphs [7, 36] even though the boundary between the two is not sharp. For example, a fact graph may have nodes representing specific real world entities such as *Joe Biden, United States*, and the edges such as *president of*. Automatically extracting ontology graphs from text [5, 24] is considerably more complex than extracting fact graphs from text for, at least, three different reasons. First, it is unclear how to configure a self-supervised ontology learning pipeline to give high performance on this task. Second, there is limited naturally occurring training data available about relationships between classes that makes it difficult to even build a supervised learning system. Finally, as we see in the above example, quite often the classes and relationships that are desirable for extraction are not directly mentioned in natural language.

Comprehensive and precise ontology graphs are important for applications in education [8], law [25], and finance [4]. As a concrete example, ontology graphs when used in conjunction with an intelligent textbook lead to significant educational improvements [8]. (In the Appendix A, we include an illustration of an existing intelligent textbook.) The requirements of this application are different from the requirements of related tasks such as answering multiple choice exam questions [13]. For example, an intelligent textbook glossary page for a class shows its super classes and sub classes. In a multiple choice exam, there are likely to be only a few questions about this relationship, and most certainly, correctly knowing all such relationships is not required. In contrast, when a student reads the intelligent textbook glossary, the subclass and super class relationships need to be all present and correct, and hence the need for an accurate and comprehensive ontology graph.

In this paper, we consider a three part process for bootstrapping the creation of accurate and comprehensive ontology graphs from textbook content. First, to identify the terms that will form the nodes of the ontology graph, we train BERT in a self-supervised manner for the task of automated term extraction, and obtain an F1 score of 0.51 as compared to an F1 score of 0.32 obtained using AWS Comprehend key phrase extraction. Second, to organize the terms into a class hierarchy, we train BERT using a weak supervision approach on the task of taxonomy learning, and obtain an F1 score of 0.55 which is substantially better than the highest F1 score of 0.3947 reported by previous methods [6]. Third, to identify the connections between the nodes in the graph, we train BERT using weak supervision for the relation extraction for the *has part* relation, and obtain an F1 score of 0.22. Even though this score is lower than the performance of previously reported methods [5, 24], the previous methods had no way to scale. In contrast, the labeling approach that we used has an accuracy of 0.96 and can be deployed at no cost on a textbook reading platform with millions of students making it possible to train and scale it to large libraries of textbooks. In summary, language models trained on textbooks provide a viable output to bootstrap term extraction, and taxonomy construction, but require more training to be viable for extracting relations such as *has part*.

We begin by describing our approach to constructing ontology graphs, then describe its experimental evaluation, and conclude by highlighting how this approach improves upon the prior work and directions for future work.

## 2. Overview of the Approach

Creating an ontology graph involves identifying classes and the relationships between them. Towards this goal, we leverage entity and relation extraction interspersed with human validation as shown in Figure 2. The components shown in the solid box on the left side of Figure 2 are part of the current system being reported in this paper, whereas the components shown in the right hand side in the dotted box, and the dotted feedback arrows are future work. We next explain the background and rationale for our approach.

A concrete target for the required ontology graph is a knowledge base, called KB Bio 101 [9], which was manually curated by domain experts and was found extremely useful in an intelligent textbook [8]. Class definitions in KB 101 used a rule language [10]. We focused our initial work on identifying direct class level relationships leaving the task of extracting their complete logical definitions for future work.

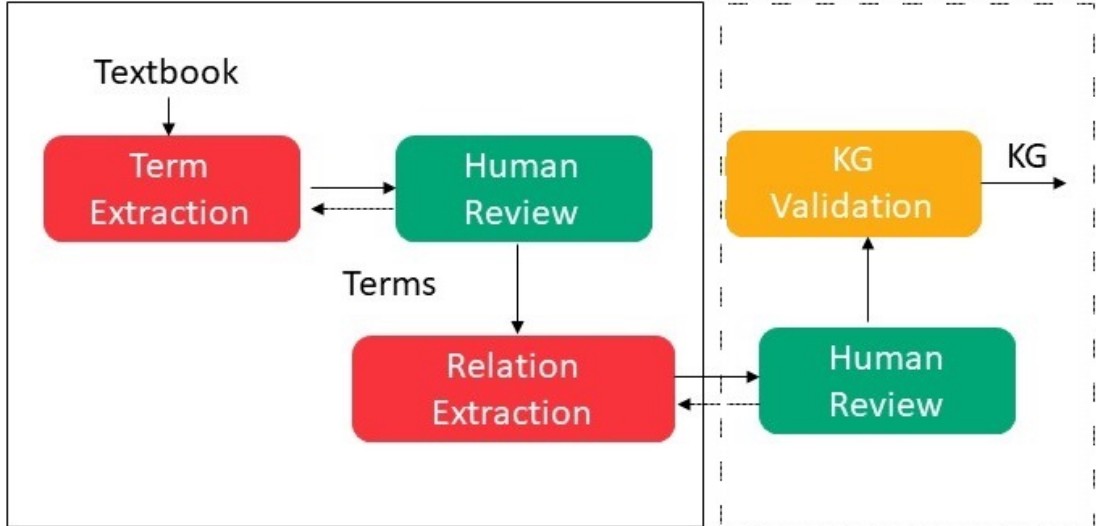

Figure 1: Ontology Graph Construction Pipeline

Automated Term Extraction (ATE) is a heavily researched task [21] which is a natural candidate for identifying the classes that should be captured in an ontology graph. Our initial experiments showed that commercial tools (for example, AWS Comprehend key phrase extraction service [1]) had a low precision, and hence, an approach that can learn the peculiarities of different domains was required. As textbooks come with a hand-curated glossary, the terms in the glossary can provide the training data necessary for automatic extractors. ATE is different from the closely related Named entity recognition (NER) task [21] that extracts concrete entities and labels them with a small number of classes such as person, location, company etc.

The choice of relationships to be represented in our ontology graph is based on the empirical evidence from KB Bio 101 and recent work on linguistic analysis [16]. We retained 20 of over 100 relationships used in KB Bio 101 that were used most often, and dropped the ones that were found confusing (e.g., *object*, *base*, etc.) or infrequently used. The relationships we currently support include taxonomic relationships for classes and instances; structural relationships such as *has part*, *has region* and *material*, spatial relationships such as *is inside* and *is above*, functional relationships such as *has function* and *facilitates*, participant relationships such as *raw material* and *result*, event structure relationships, such as *subevent* and *next event* and causal relationships such as *enables* and *prevents*. We also allow the possibility that no direct relationship may exist between two terms.

Automatic relation extraction (ARE) has been used for extracting relationships from text [7, 36, 5, 24]. We used the data from KB Bio 101 and weak labeling methods for training BERT for the relation extraction task. In the experiments that we report here, the resulting model did not have a high accuracy at this task. The primary bottleneck in improving ARE is lack of training data. Traditional approaches to training data development have relied on custom labeling tasks [5, 24]. We wanted to develop a relationship labeling task that provided a cost-effective way of gathering relationship labels, but also had the potential

to be pedagogically useful to justify no cost deployment on a textbook reading platform. Therefore, we designed a relationship selection task (RST) in which a student studying from a textbook engages in an educationally useful concept mapping activity [17, 35] and chooses the correct relationship between two classes appearing in the textbook. We plan to deploy the RST on the OpenStax open-source textbook reading platform that attracts several million students every year.

The relation extraction step, shown in Figure 2, encapsulates both RST and ARE. In the early stages, we envision the RST to play a bigger role in creating the ontology graph, but as additional data accumulate, ARE would become more predominant. The human review step for both term and relation extraction refers to a review by an expert to ensure a highly accurate ontology graph. The backward arrows indicate that as human experts perform validation, the resulting information can be fed as training data to further improve the automated extraction. Future versions of the system will include an active learning component [33] which would seek human input only for those cases where the confidence in the automated extraction is low. Knowledge validation refers to the step of verifying the ontology graph that connects different class level relationships into a global whole.

For the rest of the paper, we will focus on describing the ATE, ARE and RST modules. In the current system, the relationship extraction is performed primarily through RST. Our ARE results establish a baseline performance which we hope to improve in our future work by leveraging the data collected through RST.

## 3. Tool Development

In this section, we will describe the design and implementation of the ATE, ARE, and the RST components of the ontology graph construction pipeline.

### 3.1 Automated Term Extraction

We formulate the term extraction problem as a supervised learning task where input is a textbook sentence, and the output is a label for each word based on the BIOES labeling scheme [28] indicating which of the word sequences constitute a term.

- Input:   ['All', 'cells', 'have', 'a', 'cell', 'membrane', '.']

- Output: [ 'O', 'S', 'O', 'O', 'B', 'E', 'O']

In the above sentence 'All' is labeled by 'O' which denotes other, 'cells' is labeled by 'S' which denotes a single word term, 'cell' is labeled by 'B' indicating the beginning of a term, and 'membrane' is labeled by 'E', indicating the ending of a term. ('I' is used to label a word that is in a term, but neither its beginning or the ending word.)

We fine tuned the cased base BERT model [14] for term extraction. Our fine-tuning involved first training BERT on the textbook corpus, and then adding a fully connected softmax output layer that produces a probability distribution across the five BIOES tags for each token. The tag for a token is then the prediction with the highest probability. We undertook our implementation using Pytorch [27], huggingface transformers bert_base_cased implementation of BERT [41], used commodity hardware, and optimized using mini-batch gradient descent with the Adam optimizer (See Appendix B for more details).

### 3.2 Automated Relation Extraction

We focused on three of the most important relationships which also had the most training data available: taxonomic, structural and synonyms. A taxonomic relationship captures the *subclass* or *superclass* relationship between two terms. A structural relationship captures the *has part* or *has region* relationship between two terms. The *has part* relation is used for relating to physical entities, and *has region* relation is used to relate two spatial entities.

We formulate the relation extraction problem as a supervised learning task where the input is a sentence with two terms identified in the sentence by adding special tokens to denote the term boundaries with tags such as TERM1-START, TERM1-END, etc., and the output is one of the six relations (subclass, super class, has part/has region, has part of/has region of, synonym, OTHER). If the sentence contains more than two terms, we consider each possible pair of those terms as a prediction task. Here is an example of such an input-output pair.

- Input:    ['All', '[TERM1-START]', 'cells', '[TERM1-END]', 'have', 'a', '[TERM2-START]', 'cell' 'membrane', '[TERM2-END]', '.']

- Output:   `has part / has region`

We use a new variant of BERT known as BERT-EM [34], which modifies a pre-trained BERT model for supervised relation classification. In BERT-EM four new additional tokens are added to denote the start and end of both terms from the term pair in the sentence. These new tokens are shared across every sentence and thus allow a generalized representation of the locations of the two terms in a term pair to be learned across sentences. This modified input sequence is then run through a pre-trained BERT base encoder. The final encoder layer representations for the TERM1-START and TERM2-START tokens are then concatenated and fed into a linear layer that is run through a softmax to produce a probability distribution over relations. We implemented all models and training code using Pytorch and Python. We used the huggingface transformers bert_base_cased implementation of BERT [41]. We used commodity hardware and optimized using mini-batch gradient descent with the Adam optimizer (See Appendix C for more details).

### 3.3 Relationship Selection Tool

We developed a relationship selection tool to guide the user in choosing the correct relationship between a pair of terms in the context of a sentence [40]. The RST supports all twenty of the relationships that were mentioned in Section 2. The intended user of the RST is a college student. The user is first asked to read a section from the textbook and then to undergo a short training on the types of relationships. We designed the training using simple common-sense examples that new users would find easy to understand. For example, we explain the *is inside* relationship using a visual in which a cat is shown hiding inside a box. (A detailed walk through of the training is available in Appendix D.) As a concrete example, consider an example where the user is asked to relate the terms "cytoplasm" and "nucleus". The user first chooses the appropriate relationship family for the terms, including taxonomic, spatial, and component-based relationships. In this example, the correct relationship family is spatial and choosing this option takes them to a second set of options

to specify which spatial relationship is correct. They have an option to flip the order of the terms to ensure that the chosen relationship applies in the correct direction.

## 4. Experimental Setup

In this section, we consider the data that we used to test different modules. The code and data have been archived at: https://openstax.github.io/research-kg-learning/akbc-2021/

### 4.1 Data for Automated Term Extraction

We used 10 open source textbooks (as listed in Table 7 in the Appendix E) across multiple science subjects such as Physics, Chemistry, Biology, Astronomy and Anatomy. In addition, we used a textbook published by Macmillan Learning entitled LIFE biology [32]. Each of these textbooks contains a glossary. We used the entries in the glossaries as training data for the terms that need to be extracted.

We then used the glossary terms from each textbook to automatically insert the BIOES tags into the sentences of that book. We handled lexical modifications such as pluralization, capitalization, etc., by using spaCy [20] to pre-process the sentences and terms so that we could match the terms on their lemmatized forms. We also handled a few special cases such as acronyms and hyphenation by implementing custom utilities.

We used the data from eight textbooks except the OpenStax Biology 2e (Bio2e) and LIFE in the training set. To evaluate our models, domain experts hand labeled a development set consisting of two sections from Bio2e (Sections 4.2., 10.2) and a test set consisting of Chapter 39 from LIFE. Finally, we removed all sentences from the training set that contained any of the terms from the development and the test sets. The sentence and term counts for the resulting splits are shown in Table 7.

### 4.2 Data for Automated Relation Extraction

To train a model for the relation extraction task, we need training data that has sentences with term pairs tagged, and the correct relationship identified between those terms. We created a data set where sentences were extracted from two Biology textbooks: Bio2e and LIFE. We used a term list obtained by combining the glossary of Bio2e and all of the biology terms from KB Bio 101 [9]. We used spaCy to tag each sentence using these terms. Tagging was done using lemmatized forms to account for lexical variations.

To associate a relationship between the pair of terms appearing in each sentence, we employed weak supervision using Snorkel [29]. This process involves three steps. First, we defined a set of labeling functions where each labeling function takes as input a sentence-term pair, and outputs a relation label or a special ABSTAIN response indicating that the labelling function cannot make a judgement. Second, we apply these functions to each term pair appearing in the input sentences producing a set of labels. Finally, we aggregated the predictions to obtain a single label. The functionality needed for second step is provided in Snorkel. We describe the first and the third steps in greater detail.

We used three categories of labeling functions: pattern-based, term-based and distant supervision-based. The pattern-based functions are similar to the Hearst patterns [19] with

one difference — instead of operating on the input sentence, they operate on the dependency parse of the sentence produced by spaCy. For example, a *has* pattern label function labels two terms in a sentence with the has part/ has region relationship if the path between them in the dependency parse consists of a single step which is labeled as "has" or "have". The term-based functions operate purely on the syntactic form of the two terms and ignore the sentence they appear in. For example, if two terms end with the same base word but one has an additional modifier in front of it, this suggests a taxonomic relation (e.g., "eukaryotic cell"*subclass*"cell"). The distant supervision function uses the existing hand curated KB Bio 101 by looking up the relationship between the two terms in the KB. An exhaustive list of all the labeling functions we tried is available in Table 8. Snorkel reports the coverage of each labeling function that provides insight into the contribution of each labeling function to covering the training data. Total coverage of these labeling functions is approximately 66% which reinforces the value of the training data that is available, and suggesting that additional labeling functions would be required for more completely coverage.

Snorkel provides two different ways of aggregating the labels produced by the individual functions: hard labels and soft labels. We can produce a single hard label for each data instance by taking a majority vote across labelling functions. This means that the relation that was labelled by the most labelling functions becomes the label for that instance. If there is a tie or no label functions applied, then the label is ABSTAIN. For producing a soft label, we can produce a probability distribution across relations by using the label model included with Snorkel [30]. The label model learns a latent underlying confidence for each label function using the co-occurrences amongst each of the label functions produced in step 2. Label functions that are estimated to be more reliable are given more weight and those that are less reliable are given less weight. Thus when multiple label functions label the same instance, their label votes are combined based on their reliability estimates. For example, a term pair that receives a certain label from three label functions will have a higher confidence in that label than the one that receives a single label or conflicting labels. We experimented with both soft labels and hard labels and found that the soft labels outperformed the hard labels.

We split all available sentences into training, development, and test sets. The development set consists of all sentences from the cell structure and cell cycle chapters of both textbooks (Ch. 4 and 10 of Bio2e, Ch. 5 and 11 of LIFE). The test set consists of all sentences from Ch. 39 of LIFE. All term pairs in both the development and test sets were removed from the remainder of the data set which then formed the training set.

### 4.3 Data for Relationship Selection Task

As the ARE task covered only a small number of relationships, we tested the RST in conjunction with ATE. We used ATE to identify terms for Sections 4.2 and 10.2 from Bio2e, and Section 14.1 from the OpenStax Psychology 2e textbooks. As the precision and recall of the ATE is not perfect, two domain experts from each of the respective domain validated the terms. We then parsed the chosen section into individual sentences and automatically identified all the term pairs that existed in each sentence. A sentence that contained $N$ terms would have $\binom{N}{2}$ possible pairings, with each pairing considered to be a single task. Using the RST, the tasks were presented to users on a crowd sourcing platform.

Table 1: Comparison of ATE with AWS Comprehend

| | | AWS Comprehend | | | ATE | | |
|---|---|---|---|---|---|---|---|
| | Term Count | Precision | Recall | F1 | Precision | Recall | F1 |
| Bio2e Section 4.2 | 24 | 0.14 | 0.83 | 0.23 | **0.34** | **0.83** | **0.48** |
| Bio2e Section 10.2 | 47 | 0.20 | **0.87** | 0.32 | **0.35** | 0.85 | **0.49** |
| LIFE Chapter 39 | 596 | 0.30 | **0.65** | 0.41 | **0.59** | 0.51 | **0.55** |
| Overall | | 0.21 | **0.78** | 0.32 | **0.43** | 0.73 | **0.51** |

Workers have varying degrees of competency and tasks have varying degrees of difficulty. Simply ignoring these individual differences and using a majority voting scheme can, therefore, lead to significant estimation errors. To overcome these issues, we designed a denoising algorithm. Our model directly assigns a latent competence parameter for each worker as well as a latent difficulty parameter for each task. If a worker has an ability greater than the difficulty of the relationship, the model assumes that the worker is likely to know the correct relationship. By contrast, if the worker's ability is lesser than the difficulty of the relationship, they will likely not know the correct relationship. If a worker does not know the correct answer they will still guess at the correct relationship, and with some task-dependent probability will guess the correct relationship. The exact structure of our model is very similar to an item response theory model [38]. Our model examines all of the worker response data and estimates the competence level of each worker, the difficulty of each relationship, and the probability of guessing the correct relationship for each task. We fit our model to the collected response data using Markov Chain Monte Carlo (MCMC) methodology [3, 15, 37].

## 5. Results and Analysis

We now describe the results and analysis for each of the ATE, ARE and RST modules.

### 5.1 Automated Term Extraction

We compared the performance of ATE with AWS Comprehend key phrase extraction service which is a commercial product [1]. We processed the text under consideration through both AWS Comprehend and the ATE. For the terms returned by each, we performed identical processing to factor out lexical variations, and to accept those mutli-word predicted terms where at least one word appeared in the gold terms. We show our results in Table 1. ATE consistently outperforms AWS Comprehend on the F metric. In some cases, AWS comprehend has better recall, but its precision is much worse suggesting that training the language models on the textbooks considerably improved the predictions.

The most interesting outcome of the experiment is the failure analysis presented below. Such an analysis can be used to to better support the textbook authors in vocabulary rigor.

**Lexical Knowledge:** Lexical knowledge such as pluralizations, acronyms, different verb tenses, nominalizations, etc., are essential for good performance on ATE. Even though we were able to leverage some lexical knowledge in spaCy, our analysis of the failing cases

Table 2: Relation Extraction Model Comparison.

| Source | Precision | Recall | F1 |
|---|---|---|---|
| BERT-EM Hard Labels | **0.43** | 0.66 | **0.52** |
| BERT-EM Soft Labels | 0.30 | **0.76** | 0.43 |

revealed that there are numerous pieces of domain-specific lexical knowledge: chromatin vs chromatid, centromere vs centromeric. A textbook-specific linguistic resource, similar to Wordnet [26], is essential for better performance on ATE.

**General Terms:** General terms are not defined in the textbook glossary, and hence, will not be present in the training data. Consider the term "microscope". For a first-year college level textbook, we can expect the student to know the meaning of a microscope, and we need not define it in the glossary. But in our ontology graph, we may have an experimental process in which microscope is the primary instrument, and hence *microscope* needs to be represented. Such general terms need to be curated for every textbook. Alternatively, we can envision a scenario in which every textbook is accompanied by a glossary of prerequisite terms that it assumes a student to know which can be input as training data.

**Terms not directly mentioned in text:** There are several categories of terms that are not directly expressed in the natural language, and we consider two examples. The text often describes concepts without actually assigning them a name, for example, *mechanisms that ventilate the environmental side of those surfaces with air or water*, which can be simply referred to as a *ventilation mechanism*. Identifying such key terms poses a dual challenge. On the one hand, the term mentioned in the text is a long phrase which the automated extractor fails to pick up, and on the other hand, even if the extractor picks out the phrase, there is a challenge of associating a succinct name with the key term. There are many examples where the textbook describes different aspects of the same term which warrant a separate representation in the ontology graph, but it refers to them using the same term. For example, the psychology textbook first defines stress as a stimulus response mechanism, and then it defines stress as a physiological response. In both cases, it uses the word *stress* to define these two separate models which require distinct names in the ontology graph. Explicating and naming such indirectly stated concepts will continue to require some human intervention.

## 5.2 Automated Relation Extraction

We calculate precision, recall and F1 scores at the level of term pairs, that is, as long as the model predicted one of the six desired relationships between a term pair, we included the prediction in our calculations. The best baseline for our model is the KB Bi0 101 that was entirely created through human effort. The results are shown in Table 2. The results for individual relation specific classes are shown in Table 3.

The best results are for the subclass and super class relationships. F1 values of 0.47 and 0.62 with an average of 0.55 are much higher than an F1 value of 0.3947 that was previously reported for taxonomy extraction [6]. Even though the results for the *has part* are lower than the previously reported results for extracting such relations, but combining

Table 3: Relation Extraction Soft Label Results.

| Relation | Precision | Recall | F1 |
|---|---|---|---|
| subclass | 0.32 | 0.88 | 0.47 |
| super class | 0.53 | 0.76 | 0.62 |
| synonym | 0.25 | 1.00 | 0.42 |
| has part/has region | 0.15 | 0.50 | 0.22 |
| has part of/has region of | 0.17 | 0.80 | 0.28 |

rapidly improving language models with RST is a more viable approach. We outline below two natural refinements to ARE.

**Expanding labeling sources:** The primary bottleneck in improving the performance of relation extraction is the limitation of training data. There are four possible avenues for cost effectively expanding the training data. First, we can add more weak labeling functions using more patterns or distance supervision sources. Second, we could leverage Wordnet which has many concepts found in textbooks and can increase coverage of the training data. Third, Wikipedia and Wikidata [39] have a lot of information on academic subjects that could be leveraged to expand the training data. Finally, we could leverage crowd sourcing to inexpensively obtain relation labels. The input provided by humans is clearly complementary to weak labeling functions. Crowd sourcing can be used in a targeted manner to fill in the gaps where programmatic labels are lacking.

**Graph post processing and inference:** The model produces a list of triples, but it is unclear exactly how consistent this set is until they are all connected into a global graph. In doing so, graph processing techniques could potentially be employed to prune false positive relations and to infer false negative relations [11]. In this way, the relation extractor from text could act as a seed graph that could then be modified to greater accuracy.

### 5.3 Relationship Selection Task

There were a total of 192 tasks for Bio2e Section 4.2, 229 tasks for Bio2e Section 10.2, and 64 tasks for Section 14.1 of Psych 2e textbooks labeled by the crowd workers. From the labeled tasks, we randomly selected 50 tasks with their labels from the denoised output for each section and presented these to two subject matter experts to verify if the labels that we obtained through our experiment were, in fact, correct. We show the results in Table 4. The precision for the case when at least one of the SMEs agreed with the crowd workers was exceedingly high, i.e., 0.96 or higher. The precision if we require both SMEs to agree with the crowd workers was much lower, i.e., 0.67 or higher. We believe that with future work on better training and guidance for the expert SMEs, and refinements to the RST tool, a precision of 0.9 or higher is achievable.

### 6. Conclusions and Future Work

We presented a pipeline to create ontology graphs from textbooks by combining pre-trained language models and crowd sourcing strategies with validation by subject matter experts (SMEs). Our work is a substantial improvement over the previous baseline [18] that involved

Table 4: Validation of Relationship Selection

| Textbook section | Precision when either SME agreed | Precision when both SMEs agreed |
|---|---|---|
| Bio 4.2 | 0.98 | 0.70 |
| Bio 10.2 | 0.96 | 0.81 |
| Psych 14.1 | 0.98 | 0.67 |

SMEs investing 5 person years to engineer a knowledge base for the first 10 chapters of a biology textbook. Scaling that effort to all 56 chapters [31] would have required an estimated $1.5M budget. Additionally, the constructed KB and software were proprietary and unavailable to other researchers. Our method significantly cuts down the time and resource costs for this task and our innovations will be released in the open domain.

Our work advances the most closely related prior work on creating ontology graphs from textbooks [24, 5] in three important ways. First, our labeling task is designed in a way that it can be incorporated into the textbook reading by students making it much more practical across large libraries of textbooks. Second, we handle a much larger set of relationships that include taxonomic, structural, functional and causal relationships. Finally, we are leveraging rapidly improving pre-trained language models so that similar underlying infrastructure could be used for both entity and relation extraction.

Both automated and crowd sourcing methods have inherent limitation unless they are sufficiently moderated by human expert validation. Therefore, we view the methods presented here as bootstrapping that can give us a head start as we start to develop a new ontology graph. As we gather more validated data with the help of crowd workers, we anticipate that ARE will improve to a point where it can become self-supervised. In the interim, we anticipate that we will be able to use active learning so that we rely on crowd input only when the prediction confidence is low.

Our approach here only gives us class level relationships between nodes in an ontology. An ontology contains much deeper relationships between instances. For example, a chromosome is inside a cell only when it is a part of that cell. Chromosomes can be removed from a cell, and in that case, the *is inside* relationship does not hold. An ontology is able to capture such nuances through a proper use of quantification in rule definitions. Automatically capturing rules is an avenue for future work.

Finally, there are many applications in law and finance that are similar to textbooks that require comprehensive and precise ontology graphs. We therefore, argue, that the innovative combinations of automation and human methods in which the precision and coverage are held to a high bar is wide open area for exploration for automated knowledge base construction.

### Acknowledgment

This work was funded by the Convergence Accelerator program of the National Science Foundation and AWS Cloud Credits for Research program. We would like to acknowledge the subject matter experts in Biology (Sandra Adams, Yan Gong and Shizuka Yamada) and Psychology (Karen Watson and Kathleen Hughes) for their participation in the human validation of the terms and crowd sourced relationship data.

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

## Appendix A. Intelligent Textbook use case for an Accurate and Comprehensive Ontology

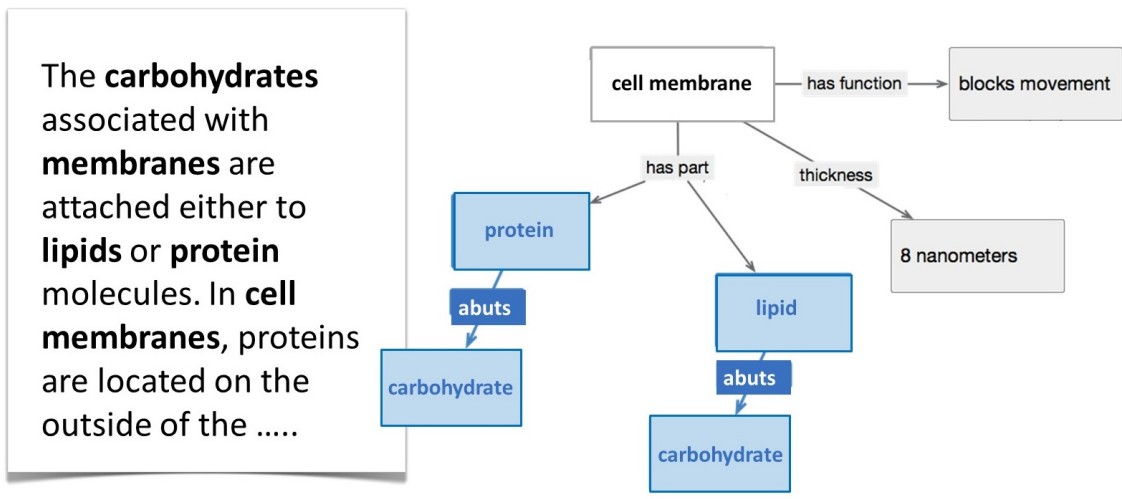

Figure 2: An example sentence from the Biology textbook and its corresponding Ontology Graph. Currently, such KGs are authored manually.

An intelligent textbook (ITB) incorporates an ontology of key concepts and relations [8] to offer five new capabilities over and above the traditional textbooks (See Figure 2 for an illustration of the ontology). First, a student can access the glossary definition of each word by simply touching on it (See Figure 3). Second, each concept has a summary page that organizes key facts about it from across the book. This includes cross linking to different diagrams and passages across the book, and thus, breaking the linear structure of the textbook. Third, it presents the visualization of each concept that a student can interactively explore. Fourth, in response to highlights in passages, it calculates relevant questions that a student can use for self-testing (See Figure 3). Fifth, it provides a structured query interface through which a student can pose questions to the textbook and obtain answers (See Figure 4). A video overview of the features is available online at: https://www.youtube.com/watch?v=I8swXc3WH1M

An ITB requires a comprehensive and accurate ontology graph for the following reasons. First, the ITB provides definition and visualization for every single concept, and therefore, the ontology graph must cover all salient concepts, their definitions, and properties. Second, the textbooks are expected to be nearly 100% accurate, and the students expect a similar accuracy from the anicillary resources such as test questions, visualizations, hyperlinks, etc. Finally, as an ITB is envisioned to be eventually a good tutor for the students, it must have an accurate and comprehensive understanding of the key concepts and relationships in the textbook comparable to what a human tutor would have.

The fluid mosaic model depicts proteins as noncovalently embedded in the phospholipid bilayer by their hydrophobic regions (or domains) or tethered to lipids inserted into the membrane. Proteins may span the membrane or may be bound on the surface. Their hydrophilic regions are exposed to the watery conditions on either side of the bilayer. Membrane proteins have several functions, including moving materials through the membrane and receiving chemical signals from the cell's external environment. Each membrane has a set of proteins suitable for the specialized functions of the cell or organelle it surrounds.

The carbohydrates associated with membranes are attached either to the lipids or to protein molecules. In cell membranes, carbohydrates are located on the outside of the cell, where they may interact with substances in the external environment. Like some of the membrane proteins, carbohydrates are crucial in recognizing specific molecules, such as those on the surfaces of adjacent cells.

Although the fluid mosaic model is largely valid for membrane structure, it does not say much about membrane composition. As you read about the various molecules in membranes in the next sections, keep in mind that some membranes have more protein than lipids, others are lipid-rich, others have significant amounts of cholesterol or other sterols, and still others are rich in carbohydrates.

What is the structure of an ER membrane?

What is the structure of a cell wall?

What are the types of Lipid?

What is the structure of lipid?

What is the structure of a carbohydrate?

NOTES      QUESTIONS

Figure 3: Textbook interface for the student. Each significant word is linked to its definition and a summary page about it. In response to student highlights, the textbook automatically generates questions from the KG that a student can use for self-testing. (Text and figures from LIFE (11th Edition) by David E. Sadava, David M. Hillis, H. Craig Heller and Sally D. Hacker. Copyright ©2017 by Macmillan Learning, Inc. Reprinted (used) by permission of Macmillan Learning, Inc.)

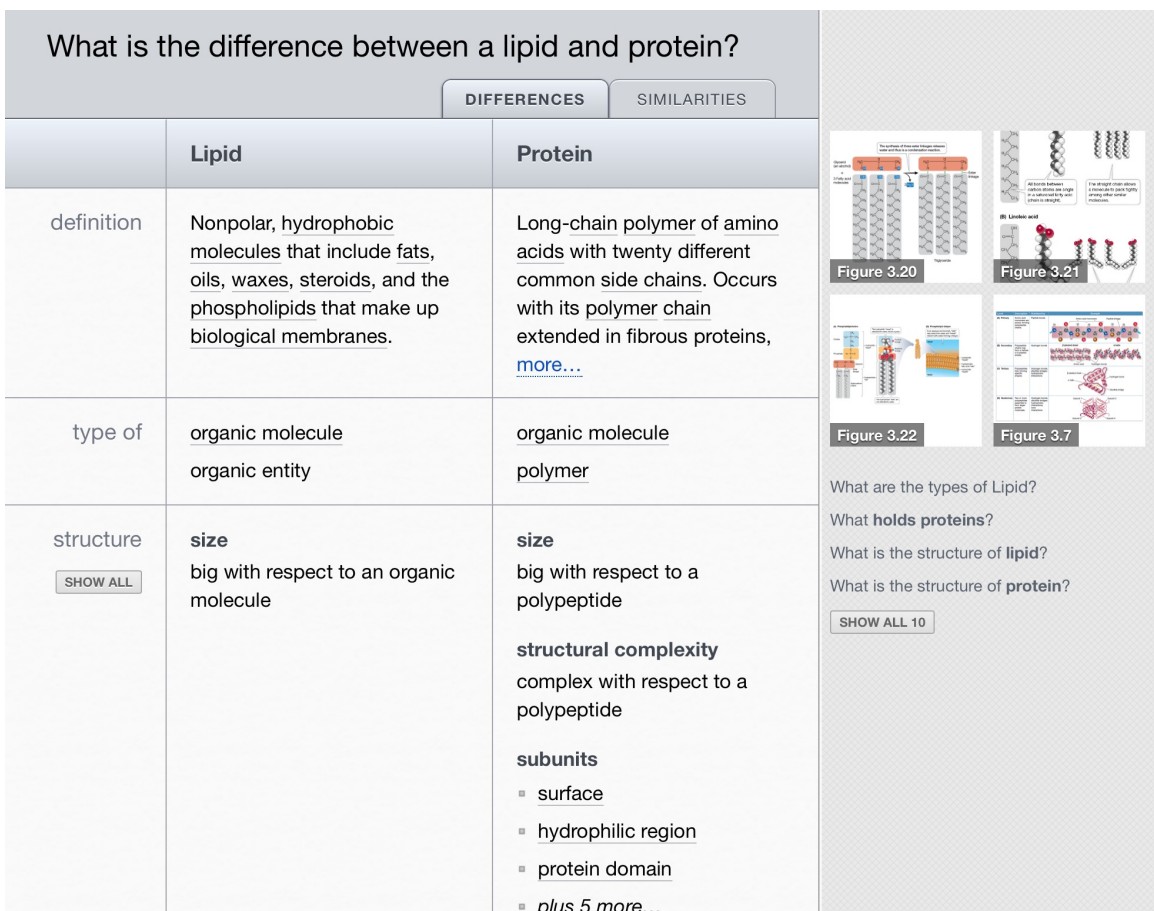

Figure 4: Through a structured query interface, a student can ask questions and the text-book returns the answers. In the above example, the student has asked to compare a protein with a lipid. (Text and figures from LIFE (11th Edition) by David E. Sadava, David M. Hillis, H. Craig Heller and Sally D. Hacker. Copyright ©2017 by Macmillan Learning, Inc. Reprinted (used) by permission of Macmillan Learning, Inc.)

An ITB of the sort illustrated in Figures 2-4 has proven its usefulness to support student learning in multiple experiments. In a study with community college, the students using the ITB scored a full letter grade higher than the students using either a paper textbook or a pure textbook. None of the students in the ITB group received a D or an F, suggesting that it may be especially helpful for low performing students. A second study in a Biology classroom at Stockholm University, investigated students interaction and learning with ITB in comparison with a traditional E-book [22]. The study showed that asking questions with the AI book was associated with higher retention and correlated positively with viewing visual representations more often. In a third study at Harvard University, the students used an ITB over a period of 1.5-months in a biology classroom [23]. It was found that students were engaged in meaningful learning throughout the study, there was a positive correlation between the use of suggested questions and the improvements in learning, and more than half the students expressed favorable opinions of the book.

## Appendix B. Optimization of the Term Extractor Model

We implemented the term extractor using Pytorch [27]. We used the huggingface transformers bert_base_cased implementation of BERT [2]. Training was performed on a single GPU available on a p2x.large on Amazon Web Services instance configured with the Ubuntu 16.04 Deep Learning AMI.

Table 5: Term Extraction Model Hyper Parameters

| Parameter | Description | Value | Search Range |
|---|---|---|---|
| # Epochs | # of passes through the data for training | 1 | N/A |
| Batch Size | # of sentences in each batch | 16 | N/A |
| Learning Rate | # learning rate for Adam optimizer | 3e-5 | [1e-5, 3e-5, 5e-5] |
| Dropout Rate | # dropout rate for output layer of BERT | 0.3 | N/A |
| Max Sentence Length | Max # of tokens in a sentence | 256 | N/A |
| Balance Loss | Whether to penalize the loss function higher for missed term labels | True | [True, False] |

We used cross entropy loss as the loss function and minimized the average negative log likelihood loss per batch. The cross entropy loss was optionally weighted to account for class imbalance giving higher penalty to missing a term label in order to bias the model towards labelling terms. Using a weighted loss function increased recall at the expense of precision and vice versa. A comparison between the two revealed that weighting the loss function performed much better overall.

Optimization was performed using mini-batch gradient descent with the Adam optimizer. We used the custom implementation of the Adam optimizer packaged with the transformers library. We used the default parameters for Adam except that we opted not to correct the bias as in the original BERT paper. We searched over learning rates in [1e-5,

Table 6: Relation Extraction Model Hyper Parameters

| Parameter | Description | Value |
|---|---|---|
| # Epochs | # of passes through the data for training | 2 |
| Batch Size | # of sentences in each batch | 16 |
| Learning Rate | # learning rate for Adam optimizer | 5e-6 |
| Max Sentence Length | Max # of tokens in a sentence | 256 |

3e-5, and 5e-5] as recommended in the BERT paper. Early stopping was utilized which halts optimization if validation loss does not improve on the next epoch.

In Table 5, we list the hyper parameter settings for the model that performed best on the development set. An exhaustive search was not performed, instead we only tried varying the learning rate and whether the loss was balanced. All of the BERT parameters were not varied but an additional dropout layer was added to the BERT output layer prior to the classifier layer.

## Appendix C. Optimization of the Relation Extractor Model

All models and training code were implemented using Pytorch and Python. We used the huggingface transformers bert_base_cased implementation of BERT [41]. Training was performed on a single GPU available on a p2x.large Amazon Web Services instance configured with the Ubuntu 16.04 Deep Learning AMI.

We used a cross entropy loss function and minimized the average negative log likelihood loss per batch. The loss function is the same with both the soft and hard labels, but the target distribution for the cross entropy loss was different. For the hard labels, it was 0 for all classes except the true label. For the soft labels, the target distribution was the probability distribution provided by the soft labels.

Optimization was performed using mini-batch gradient descent with the Adam optimizer. Early stopping was utilized which halted the optimization after the validation metrics had not improved an epoch. In Table 6, we list the most important hyper parameters and the settings when best observed performance on the development set was achieved.

## Appendix D. Relation Selection Tool Walk-through

A video demonstration of the RST is available at: https://www.youtube.com/watch?v=bs5U1M6tgBg

### Relationship selection: Training

In the next portion of this study, you will be identifying how the different biology concepts you just read about are related. Before you begin, we will walk you through a brief training.

During the relationships selection task, you will be shown two terms. Your job will be to identify the relationship that exists between the terms (if there is one).

There are a LOT of possible relationships available and so, to make this easier, we have broken the task up into two parts:

- First, you will select a high-level relationship family
- Second, you will select the specific relationship within the family that you have chosen

DON'T FEEL LIKE YOU NEED TO MEMORIZE ALL THE RELATIONSHIPS! There will be reminders in the task to help you if you get stuck.

Figure 5: This is an opening screen in which the user is introduced to the task

### Entities vs Events

There are two kinds of terms that you will link: Entities and Events.

- Entities correspond to actual things (like cats, tables, cells, etc)
- Events correspond to things that happen (like falling, starting a car, mitosis, etc)

Figure 6: We first introduce the user to the distinction between an event and an entity

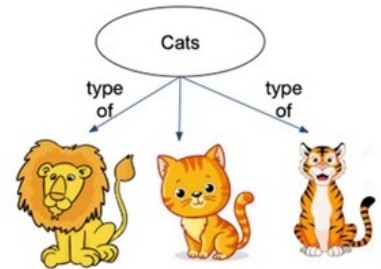
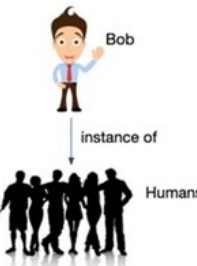

Figure 7: We teach the user about taxonomic relationships

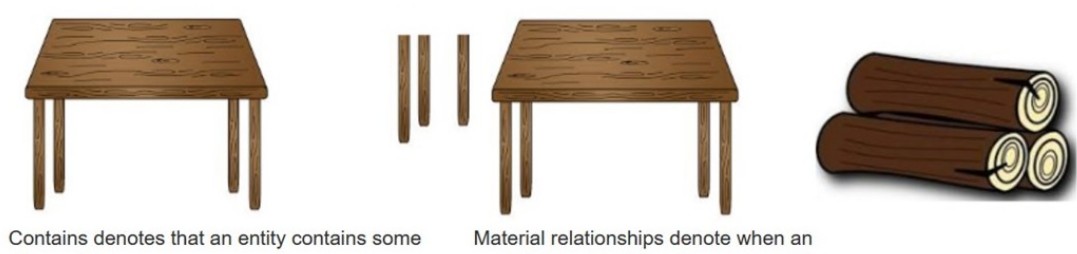

Figure 8: This example illustrates component relationships

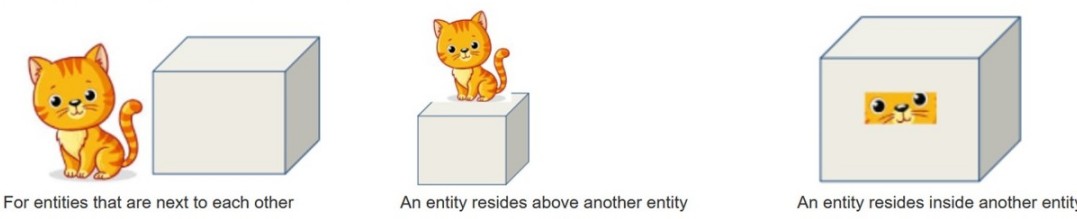

Figure 9: Example illustrations for spatial relationships

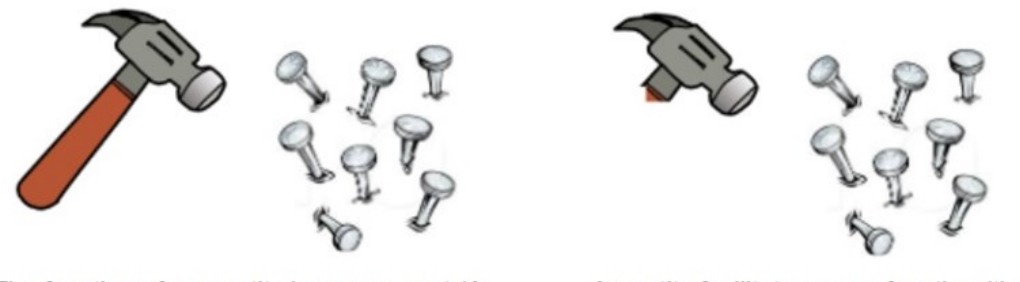

## Functional Relationships (Entity to Event)

These relationships describe how entities are located around other entities

The function of one entity is some event (A hammer nails things)

An entity facilitates some function (the hammerhead facilitates the nailing)

Figure 10: Examples to illustrate functional relationships

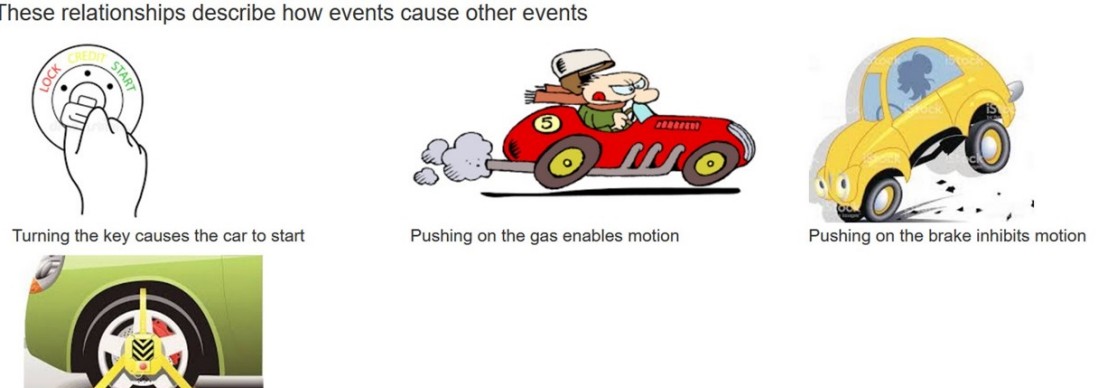

## Causal Relationships (Event to Event)

These relationships describe how events cause other events

Turning the key causes the car to start

Pushing on the gas enables motion

Pushing on the brake inhibits motion

A boot on the tire prevents motion

Figure 11: Examples to illustrate causal relationships

## Event Structure Relationships (Event to Event)

These relationships describe when events occur with respect to other events.

Consider the example of how to boil potatoes:

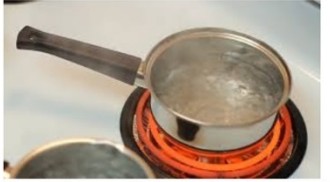 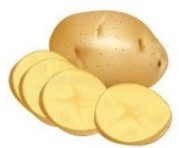 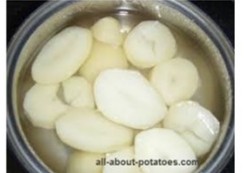

Step 1: Put pot of water on stove to boil (first event)

Step 2: While the water boils, cut up the potatoes (subevent)

Step 3: After the water boils, add the potatoes (next event)

Figure 12: Examples to illustrate event structure relationships

## Participant Relationships (Entity to Event)

These relationships describe how entities participate in events.

The cat pushed the vase with it's paw. The vase broke on the ground into many pieces.

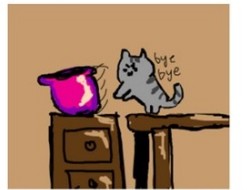 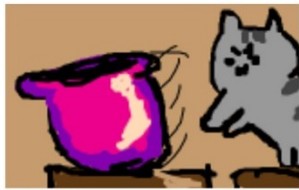 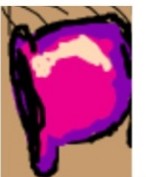

The cat pushed the vase (participants)

The cat used it's paw (instrument)

The vase landed on the ground (site)

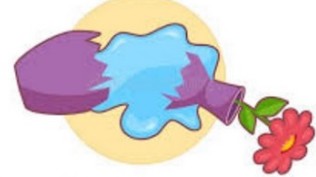

The vase broke into many pieces (result)

Figure 13: Examples to illustrate participant relationships

## No Direct Relationships

Often (maybe most of the time) terms won't have a direct relationship.

Ex: The cat is on the table. The dog is outside ==> the cat and dog have no direct relationship.

Ex: The table has legs. The legs are plastic ==> the table and plastic are related through the legs, but not directly related to each other

Figure 14: Introducing the possibility of no direct relationship

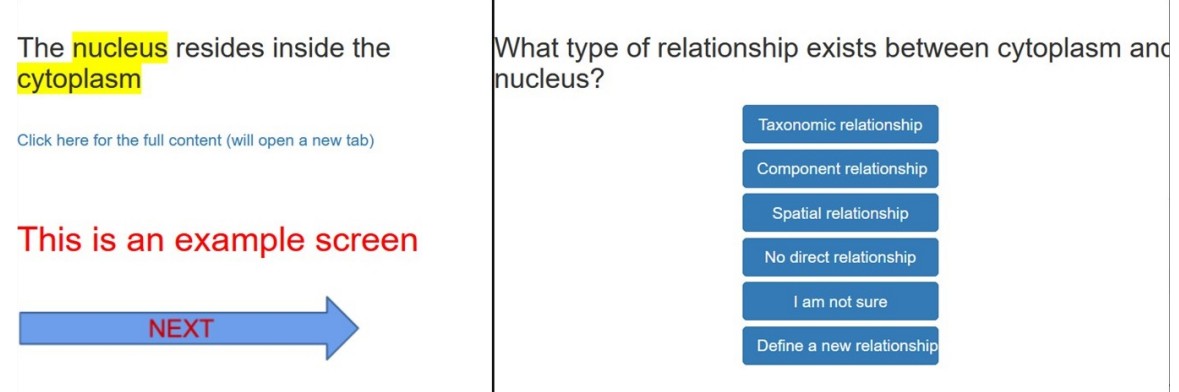

Figure 15: Start of the dialog for relation selection

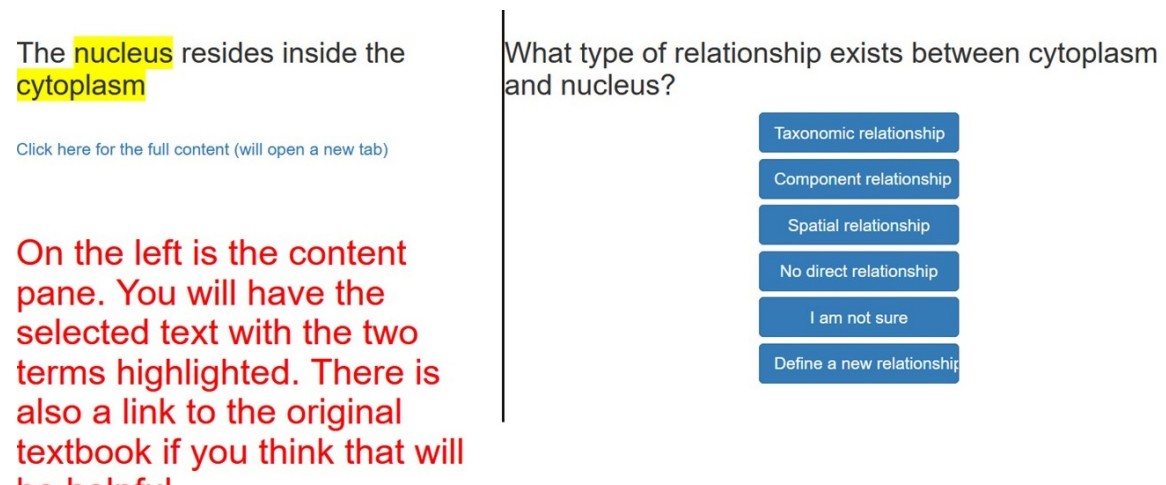

Figure 16: Explanation for the relation selection screen

The nucleus resides inside the cytoplasm

Click here for the full content (will open a new tab)

On the right is the relationship selection pane. You can click on one of the buttons to choose a relationship family or (later) a specific relationship. You can hover your mouse over any of the buttons to get a reminder of what each relationship entails.

What type of relationship exists between cytoplasm and nucleus?

Taxonomic relationship
Component relationship
Spatial relationship
No direct relationship
I am not sure
Define a new relationship

Figure 17: Continued explanation for the relation selection screen

The nucleus resides inside the cytoplasm

Click here for the full content (will open a new tab)

Our first task is to pick the relationship family that relates these terms. Since the text talks about the nucleus being inside the cytoplasm we should pick the 'spatial' relationship family

What type of relationship exists between cytoplasm nucleus?

Taxonomic relationship
Component relationship
Spatial relationship
No direct relationship
I am not sure
Define a new relationship

Figure 18: The user is instructed to first choose the relationship family

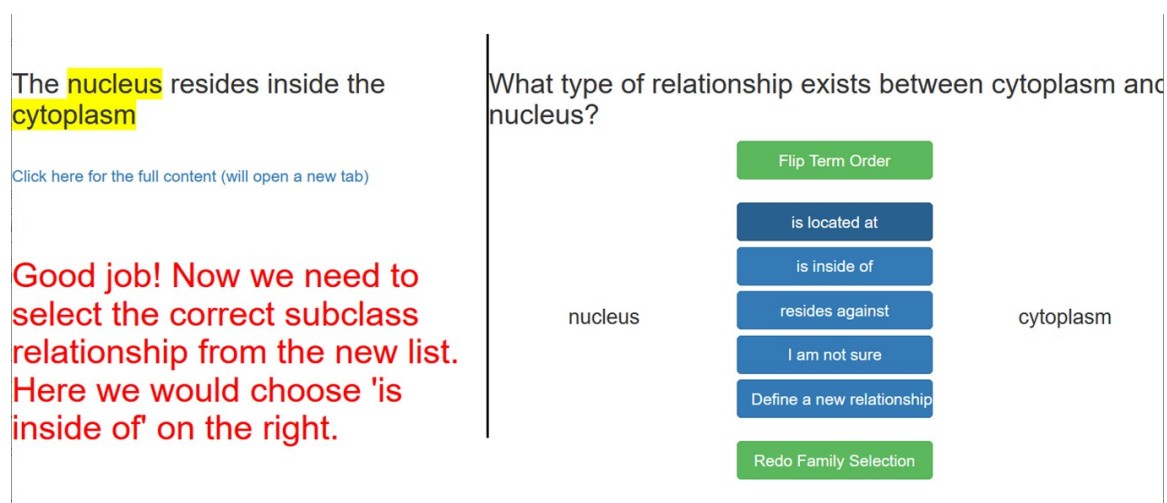

Figure 19: Once the user correctly chooses the relationship family, they are instructed to choose the relationship

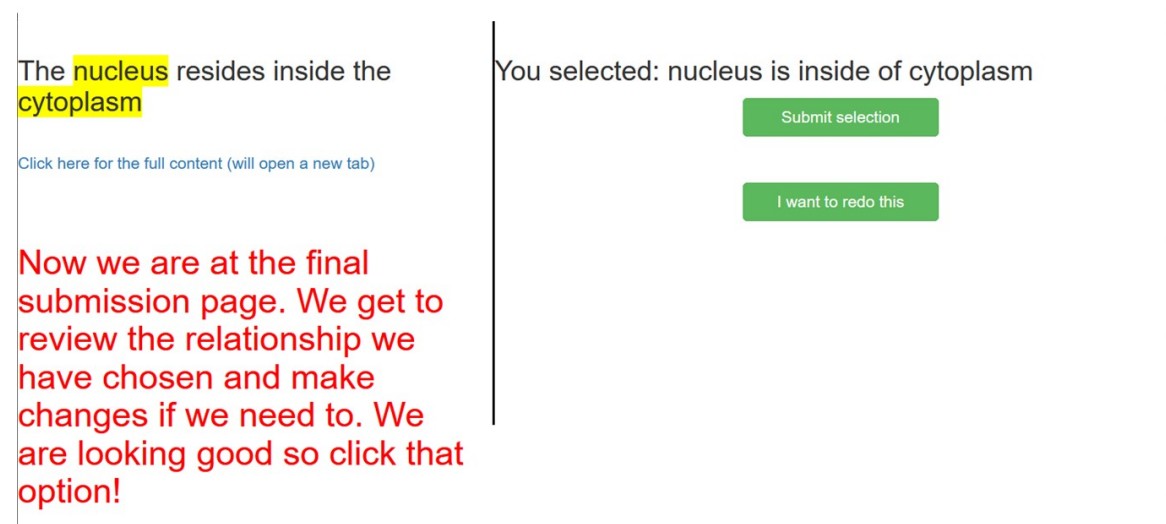

Figure 20: The user is asked to confirm the choice of relationship

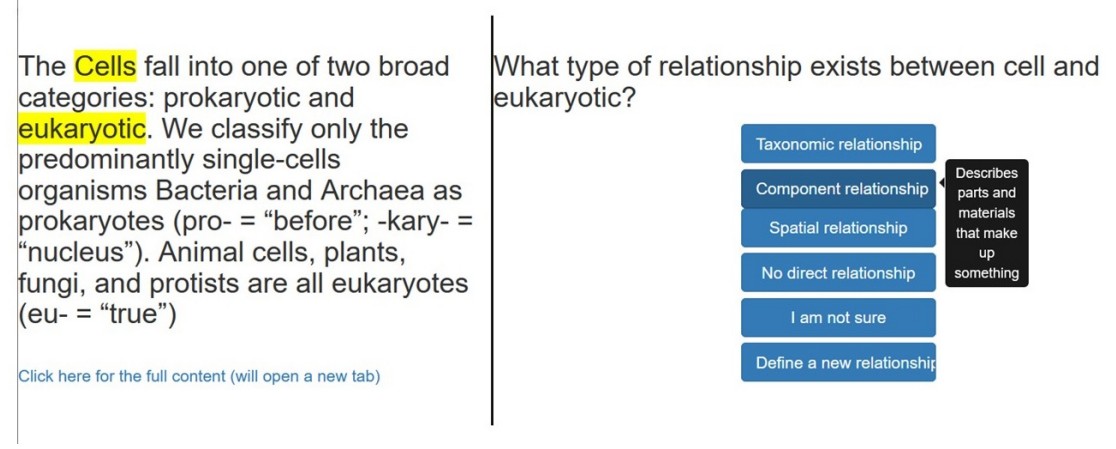

Figure 21: Another example of a relationship selection task

Like prokaryotes, eukaryotic cells have a plasma membrane, a phospholipid bilayer with embedded proteins that separates the internal contents of the cell from its surrounding environment.

Click here for the full content (will open a new tab)

**Excellent job! While there definitely are connections between eukaryotic cells and proteins, here proteins refers to a component of the plasma membrane. Hence,**

What type of relationship exists between eukaryotic cells and protein?

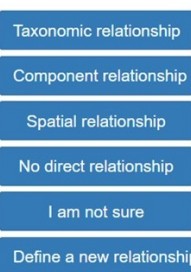

Figure 22: An example where the relationship between the terms may not be direct

You have finished the training modules!

You will now be moved on to create your own relationships on new content.

You will spend 30 minutes on these tasks. There is a timer in the upper right hand corner to show you how long you have spent thus far.

Remember that you can always hover your mouse over the relationship choices to get a reminder on what they mean.

Finally, you can always chose the 'I don't know' button to get a new example if you get stuck on one example.

Figure 23: Confirmation of the completion of the training

Animal cells, plants, fungi, and protists are all eukaryotes (eu- = "true").

Click here for the full content (will open a new tab)

What type of relationship exists between eukaryotes an protists?

Taxonomic relationship

Component relationship

Spatial relationship

No direct relationship

I am not sure

Define a new relationship

Figure 24: An example relationship selection task after the user has completed the training

## Appendix E. Textbook Dataset

Table 7: Textbook Dataset

| Textbook | # Sentences | # Terms |
|---|---|---|
| OpenStax Anatomy & Physiology | 21706 | 3196 |
| OpenStax Astronomy | 18844 | 810 |
| OpenStax Biology 2e | 24544 | 2757 |
| OpenStax Chemistry 2e | 13799 | 954 |
| OpenStax Microbiology | 16190 | 4149 |
| OpenStax Psychology | 9967 | 1086 |
| OpenStax Physics Volume I | 15005 | 462 |
| OpenStax Physics Volume II | 11779 | 466 |
| OpenStax Physics Volume III | 9250 | 580 |
| LIFE Biology | 16673 | 2305 |

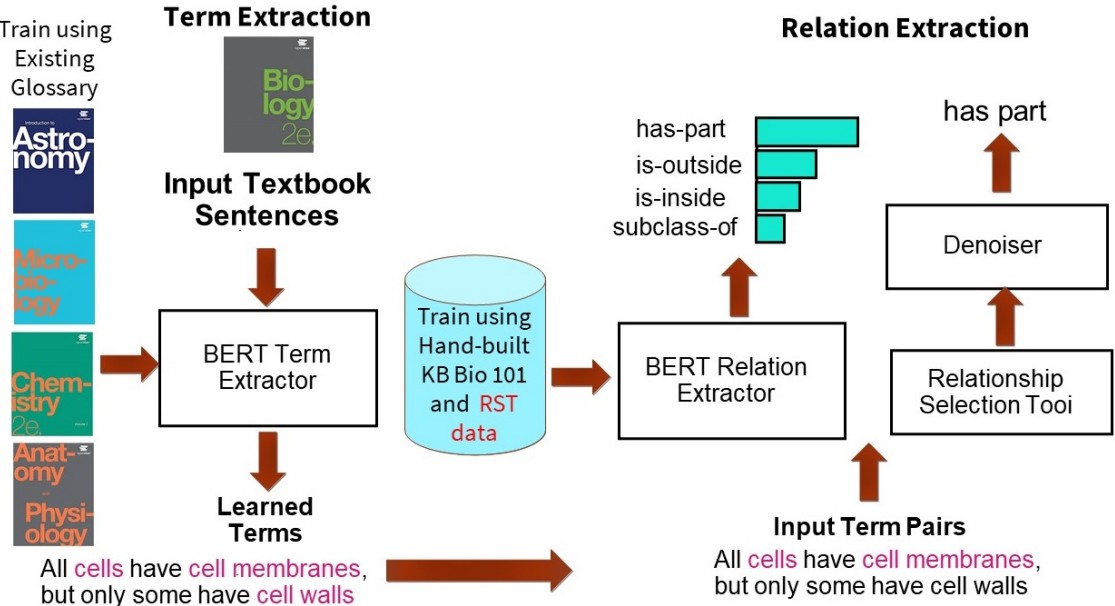

Figure 25: Illustration of the use of data in the Ontology Graph construction pipeline. Recall that the RST data will be used for training the ARE in the future versions of the system.

## Appendix F. Performance of Labeling Functions

Table 8: Label Functions.

| Label Function | Description | Relations | Coverage |
|---|---|---|---|
| Pattern-Based | | | |
| Has | X have/has Y | HAS-PART/REGION | 0.4% |
| In | X in the/a/an Y | PART/REGION-OF | 0.6% |
| Possessive | X's Y | HAS-PART/REGION | 0.2% |
| Contains | X contains/containing Y | HAS-PART/REGION | 0.2% |
| Consist | X consists/consisting of Y | HAS-PART/REGION | $\leq 0.1\%$ |
| Part Of | X is/are part(s) of | PART/REGION-OF | $\leq 0.1\%$ |
| Is A | X is a/an Y | SUBCLASS | 0.5% |
| Such As | X such as Y | SUPERCLASS | 0.5% |
| Including | X including Y | SUPERCLASS | 0.1% |
| Called | X called Y | SUPERCLASS | 0.2% |
| Especially | X especially Y | SUPERCLASS | $\leq 0.1\%$ |
| Appos | X, a/an Y | SUBCLASS | 0.1% |
| And Other | X and/or other Y | SUBCLASS | 0.1% |
| Are | X are Y | SUBCLASS | 0.4% |
| Symbol | X[,-]Y[,-] | SUPERCLASS | 0.2% |
| Also Known | X also known as Y | SYNONYM | $\leq 0.1\%$ |
| Also Called | X also called Y | SYNONYM | $\leq 0.1\%$ |
| Parens | X (Y) | SYNONYM | 0.3% |
| Plural | X (plural/singular = Y) | SYNONYM | $\leq 0.1\%$ |
| Term-Based | | | |
| Term Modifier | eukaryotic cell, cell | SUBCLASS, SUPERCLASS | 1% |
| Term Subset | oncogene gene | SUBCLASS, SUPERCLASS | 0.2% |
| Distant Supervision KB Bio 101 | | | |
| other | no relation in KB | SYNONYM | 66% |
| has-part | has-part in KB | HAS-PART/REGION | 1.4% |
| has-region | has-region in the KB | HAS-PART/REGION | 0.3% |
| subclass | subclass in the KB | SUBCLASS | 1.3% |
| synonym | synonym in the KB | SYNONYM | 0.3% |