# OpenReview forum: "A Case Study in Bootstrapping Ontology Graphs from Textbooks"
_AKBC.ws/2021/Conference — AKBC 2021_

### Official Review · Reviewer_49XT · 2021-07-22
**This paper proposes to construct ontology graph by adapting BERT and leveraging a novel crowd sourced relationship selection task, which shows that this approach is able to bootstrap the ontology graph creation for further refinement and improvement through human effort.**

**Rating:** 7
**Confidence:** 4

**Review:**

Strength:
1. The labeling task is designed for the textbook reading by students across large libraries of textbooks to obtain plenty of data for relation selection.
2. The framework can handle diverse relationships that include taxonomic, structural, functional and causal relationships.
3. This paper also considers SOTA BERT-based model for sub-tasks .


Weekness:

The paper mentions active learning will be considered later, which indeed is a good next step for this work.

One related work can be cited.
"Constructing Taxonomies from Pretrained Language Models, Chen et al, NAACL 2021"

---

> ### Author Response · Authors · 2021-07-30
> **Restriction to a tree-structured taxonomy is a big difference**
>
> Thanks for pointing out the useful related work: Constructing Taxonomies from Pretrained Language Models, Chen et al, NAACL 2021". The approach used by Chen et. al. is very similar to our work in that we are leveraging BERT for relationship prediction. Here are some interesting differences:
>
> 1) Chen et. al. begin with a list of terms. One challenge in our work was that the list of terms itself had to be extracted from the text.
>
> 2) Chen et. al. assume that the target taxonomy is a tree, which is an artificial restriction. The Wordnet hyponym hierarchy is not a tree. Similarly, the taxonomy considered in our paper is not a tree. Therefore, the Tree Reconciliation step used in Chen et. al. is not appropriate for our work.
>
> 3) As explained in the Section 5.1 of the paper, all the terms that are of interest for an ontology are not always mentioned explicitly in the text making the problem of inferring relationships between such concepts more difficult.
>
> 4) The approach used in Chen et. al. for generating negative training examples is interesting, and it could be worth exploring in our framework.
>
> 5) The approach used in Chen et. al. predicts only the taxonomy relation while we predict multiple relationships using the same model. It is an interesting question whether it might be better to train a single model for each relation vs training a combined model as we considered.

---

### Official Review · Reviewer_Y5sz · 2021-07-22
**An interesting experiment on an interesting research topic**

**Rating:** 6
**Confidence:** 4

**Review:**

Overall, I like this work. It is well written and easy to follow. It feels like there is still a fair amount of work to get to well-rounded conclusions, but this seems like a good stage to already make initial results available for the community. Here and there, the text could be more clear by only discussing what has been done already and relegating all future steps to the discussion at the end. A further major aspect that needs improvement is comparing the obtained results to baselines. My comments on several aspects are below:

One aspect which is always hard is to differentiate between instances/entities/individuals and classes/types/concepts. I think you are doing a good job at explaining that. But, I wonder whether it is really that important to distinguish the two.

You mention that 'extracting rule definitions is out of the reach of automated methods'. I think this is not completely true. I have seen presentations on methods for doing that. Perhaps they are not performing well enough for your purpose?

Selecting 20 out of the 100 available relations, based on you thinking they are 'confusing' or their low usage frequency is a weakness of your work. I would have liked to see you using these 100 relations and then gain an insight on whether what you would a priori thought to be confusing and low frequency relations, really lead to issues. This would be an interesting ablation study, rather than something to exclude from the outset.

In 3.1 you describe your model for ATE. It remains unclear to me whether you still free the weights of the BERT model while training for the prediction. If you only allow the weights of your softmax layer to be trained, you might not be taking the full advantage of the possibilities of your model.
In the same section you also mention you use 'standard techniques for optimizing'. Please provide more detail here. Small differences might have large effects. Also, I would hope you could share the full implementation of the work done as this would make it possible for others to run comparative studies.
Also, there are a lot of existing methods for this step you could compare with.

I section 3.2, you mention you use sentences with two terms identified in the sentence. I do not get why this is limited to two and sentences with more terms are not included for this.

For your experiments I would have expected more attention paid to the creation of good baselines. For example systems like NELL have been used for relation extraction before and also systems like Yago have extensive implementations for relation extraction from text. Even using a simple baseline based on Hearst patterns (which you are aware about), would be expected. Besides, there are also specific works that use distributional semantics for automatic taxonomy induction. These would be only limited to some of the relations types you consider, but would still be very relevant.

In section 3.3, I think it is very nice that you include the option of flipping the relationship in the correct direction. Even nicer would be to see whether you can use that information to train a model and do this flipping task automatically.

One limitation of your work is that (co-)references are not handled.  This is in part because you feed your models one sentence at a time.  I think there is still an interesting direction where you would give your model longer pieces of text including coreferences and see whether it can figure those out while extracting the relations.

In 4.3. You mention you employed biologists to check the correctness of the psychology text annotations as well. Did they consider annotating this data harder than annotating the biology text? Were there more inter-annotator disagreements?

In section 4.3, you describe your use of a 'sophisticated denoising algorithm' (with reference to 37 and 9). Please give a summary or at least an intuition on what that is doing in your paper.

In 3.2 and 5.2, you reduce your set further to only 6 relations, while earlier you mentioned 20. This becomes very limiting.

---

> ### Author Response · Authors · 2021-07-30
> **Many valid points, and good suggestions!**
>
> Thanks so much for your careful and considered review. Our responses are below.
>
> 1) We have reworded the overview section to more clearly delineate between the current work and the future work.
>
> 2) Whether it is necessary to distinguish between classes and individuals?
>
> We make this distinction for two reasons. First, the training data at class level is much smaller than the training data for individuals, and this reinforces the difficulty in training automated methods for relation identification. Second, as explained in Section 6, classes capture nuances in relationships through proper use of quantification. For example, a chromosome is inside a cell only when it is a part of that cell. Chromosomes can be removed from a cell, and in that case, the "is inside" relationship does not hold true. The individual level relationships cannot correctly capture such information making it necessary to distinguish between a class level and instance level graph.
>
> 3) State of the art on rule extraction.
>
> We can provide a comparison if you give us a specific citation. Most closely related rule extraction work is [5,24]. The primary advancements in our work are (a) the use of state of the art pre-trained language models that are rapidly improving, and (b) a more scalable labeling approach that can be deployed in the context of textbook reading by millions of students. Because of this a comparison to methods used in [5,24] is not meaningful.
>
> 4) Choosing 20 out of 100 relationships.
>
> Recall that the labeling tool is to be deployed in the context of an online textbook reading experience for students, and hence, the training requirement needs to be very low. The use of 100 relationships in previous projects required a training time of 20 hours  which is not feasible for data collection from students. The number of relationships dropped because of their confusing nature was only three (it included agent, object, and base). The feedback from the publishers interested in adopting this technology has been to provide them with something much less complex but which includes maximum automation.
>
> 5) We will publicly release all the code and data in the open source. As OpenStax textbooks are open source, we are hoping that this dataset will itself be of great interest to this community for future algorithm development. We have noted this in the revised version of the paper.
>
> 6)  We trained all the weights, not just the softmax layer. (Fine-tuning refers to this process instead of just training the final layer). We have clarified this in the revised version of the paper.
>
> 7) We have added Appendix D and E that provide details of the optimization and details. A summary is included in the main body of the revised paper.
>
> 8) We have clarified in Section 3.2 that if a sentence contains more than two terms, we consider all possible combinations of those terms for identifying possible relationships between those terms.
>
> 9) Neither YAGO nor NELL are appropriate comparisons for our relation extraction work for the following reasons. Both are fact extraction systems and do not extract taxonomies, and relations such as has part/has region.
>
> 10) In the SemEval 2016 evaluation, Task 13, several taxonomy extraction systems were evaluated with the best F score of 0.3947.  The distributional semantics based method had a score of of 0.24. We have added this discussion to the paper.
>
> 11) Regarding comparison to baselines such as Hearst Patterns, we have added a table in Appendix F that shows the performance of each of the extraction patterns we used and discussion added in the results section.
>
> 12) That's a worthy idea for investigating in the future, but was outside the scope of the work for this project.
>
> 13) Coreference resolution. This is another great direction for future work.
>
> 14) Section 4.3. The psychology terms were validated by the psychology domain experts.
> The paper has been corrected to reflect this.
>
> 15)  We have provided a brief overview of the denoiser in the Section 4.3 of the revised version.
>
> 16) Set of six relations.
>
> A reduced set of 6 relations was used only for the ARE experiments. This smaller subset was chosen for an initial experiment and for the relations that had the highest amount of training data available. Even for this small subset, the current ARE algorithms are not above a practically useful threshold. Therefore, it did not make sense to test with relations that have even lesser training data available.

---

### Official Review · Reviewer_a2gb · 2021-07-23
**Interesting ideas, not much science**

**Rating:** 5
**Confidence:** 5

**Review:**

I like the mission of this nearly 2-decade old project and this seems like a reasonable thing to try, an epsilon increment on the idea of populating a class-level graph for the purposes of supporting a textbook. All told, they take a bunch of existing ideas and tack them on to something that is already there, integrating feedback into the tool is a great idea in particular.

However the paper reads like a progress report or masters project summary, and comes to no conclusion about the methods other than simply offering some metric numbers.  It describes what has been done, but has no hypothesis that was tested.  For this to be science,  we'd need at the very least a comparison to some baseline, an evaluation of different approaches and comparison of their advantages and disadvantages.

---

> ### Author Response · Authors · 2021-07-30
> **Substantial progress**
>
> We thank the reviewer for the feedback. As no citation of the mentioned old project was provided, we will assume that the reviewer is referring to Project Halo. If we consider Project Halo as a baseline, the current work is a significant advancement. The developers in Project Halo used knowledge authoring by domain experts which provided a highly detailed and accurate gold standard for knowledge base content that is not available anywhere else. The present paper was framed around the engineering question: How do we bootstrap a comprehensive and precise ontology graph from text? The following specific conclusions answer this question and have been incorporated into the paper upfront in the introduction section.
>
> 1) Automated term extraction using language models is a viable technology partially to automate the Term Identification step of knowledge authoring.  This conclusion is backed by the F1 score of  0.51 on automated term extraction. In Project Halo, all of this work was done manually. A commercial off-the-shelf key phrase extractor provided by AWS produces an F score of 0.32 on this data set, and hence, our ATE offers a substantial improvement over the state of the art.
>
> 2) Automatic taxonomy construction using language models is a viable technology to support taxonomy creation. This conclusion is backed by an average F1 value of 0.55 for taxonomy construction which is substantially better than the highest F score of 0.3947 reported by taxonomy learning methods evaluated at the SemEval 2016 shared Task 13.
>
> 3) Automated relation extraction, for the relations such as has part (F1 score of 0.22) is currently not a viable technology to support knowledge authoring. The new labeling tool reported here has an accuracy of over 0.96,  can be deployed on a large textbook reading platform and will eventually produce training data for free that will help the community in significantly improving relation extraction technology.
>
> The framework reported here is radically different from Project Halo that was locked into proprietary textbooks and licensed software. All the textbook data and the extraction algorithms will be openly available for the community to freely use and develop further. Eventually a large library of knowledge graphs for all 42 textbooks published by OpenStax will be created. We are hoping that this paper will introduce an invaluable dataset to advance the research in the AKBC research community.

---

### Decision · Program_Chairs · 2021-08-18

**Decision:**

Accept

**Comment:**

This paper presents results of a project to bootstrap ontology graphs from student textbooks.

A major weakness of the approach is the limited comparison with prior work. Clearly, it is a non-trivial problem to compare such a complex system to prior work, given the number of different components involved, but more effort could have gone into it. On the plus side, the approach draws on pre-trained language models and thus shows some of the new possibilities compared to some of the work from the past.

The reviewers felt that the approach and results currently seem a bit premature, and various components could be improved. In light of this, the title of the paper should be made more specific to better describe this as a case study on specific kinds of data.

The authors have promised to make their dataset available, which should be valuable for the community.